# Research on the Detection Method of Organic Matter in Tea Garden Soil Based on Image Information and Hyperspectral Data Fusion

**DOI:** 10.3390/s23249684

**Published:** 2023-12-07

**Authors:** Haowen Zhang, Qinghai He, Chongshan Yang, Min Lu, Zhongyuan Liu, Xiaojia Zhang, Xiaoli Li, Chunwang Dong

**Affiliations:** 1Tea Research Institute, Shandong Academy of Agricultural Sciences, Jinan 250100, China; 2022120748@sdau.edu.cn (H.Z.); y1029345485@email.swu.edu.cn (C.Y.); lumin@saas.ac.cn (M.L.); lzy0629@emails.bjut.edu.cn (Z.L.); zhangxiaojia@saas.ac.cn (X.Z.); 2Shandong Academy of Agricultural Machinery Science, Jinan 250100, China; 11913056@zju.edu.cn; 3College of Biosystems Engineering and Food Science, Zhejiang University, Hangzhou 310008, China; xiaolili@zju.edu.cn; 4College of Mechanical and Electrical Engineering, Shihezi University, Shihezi 832000, China

**Keywords:** hyperspectral, machine visualization properties, data fusion, tea plantation soils, organic matter

## Abstract

Soil organic matter is an important component that reflects soil fertility and promotes plant growth. The soil of typical Chinese tea plantations was used as the research object in this work, and by combining soil hyperspectral data and image texture characteristics, a quantitative prediction model of soil organic matter based on machine vision and hyperspectral imaging technology was built. Three methods, standard normalized variate (SNV), multisource scattering correction (MSC), and smoothing, were first used to preprocess the spectra. After that, random frog (RF), variable combination population analysis (VCPA), and variable combination population analysis and iterative retained information variable (VCPA-IRIV) algorithms were used to extract the characteristic bands. Finally, the quantitative prediction model of nonlinear support vector regression (SVR) and linear partial least squares regression (PLSR) for soil organic matter was established by combining nine color features and five texture features of hyperspectral images. The outcomes demonstrate that, in comparison to single spectral data, fusion data may greatly increase the performance of the prediction model, with MSC + VCPA-IRIV + SVR (R^2^C = 0.995, R^2^P = 0.986, RPD = 8.155) being the optimal approach combination. This work offers excellent justification for more investigation into nondestructive methods for determining the amount of organic matter in soil.

## 1. Introduction

Worldwide, tea plants are produced as a significant revenue crop [1]. Because of its distinct flavor and scent, tea is the most consumed nonalcoholic beverage in the world. The organic matter in the soil has a large influence on how the tea tree grows. The whole soil ecology depends heavily on soil organic matter, which also helps with nitrogen fixation, carbon sequestration, and plant nutrient retention [2]. As a result, accurate information on the amount of organic matter in the soil is crucial for enhancing tea plants’ ability to thrive. However, conventional techniques for determining soil organic matter are labor-intensive, costly, and time-consuming, and leave behind dangerous chemical residues. An example is the use of potassium dichromate–sulfuric acid solutions to oxidize soil organic carbon. The use of chemical reagents in this method can damage the physical and chemical environment of the soil itself. Time consumption and chemical residues are also drawbacks of this method. The quick detection of soil organic materials during the growth of tea trees cannot be met by this [3].

Machine vision and hyperspectral imaging are two nondestructive testing techniques that are widely utilized in many industries, including forestry [4], agriculture [5], animal husbandry [6], and fisheries [7]. Hyperspectral imaging technology is a technology based on image data in multiple narrow bands. It combines imaging technology with spectral technology to detect the two-dimensional geometric space and one-dimensional spectral information of the target and obtain continuous, narrow-band image data with high spectral resolution. Based on hyperspectral data, Yun Chen et al. [8] developed three models for soil organic matter: simple linear regression (SLR), multivariate stepwise linear regression (SMLR), and partial least squares regression (PLSR). Of these, the PLSR model proved to be the most successful. Rahimi-Ajdadi et al. [9] employed soil photographs to estimate the soil water content using feature finding in several color spaces, and their results had an average absolute error of less than 1.1%. However, data from a single source may not be perfect in defining some qualities, and individuals have begun to employ fused data in the field of nondestructive testing to further enhance target identification. Ting An et al. [10] collected visual (color pictures) and olfactory (sensor array spectra) data on tea leaves during the fermentation of black tea and combined them with hyperspectral data to create a discriminative model for the level of fermentation that had a 95% prediction set accuracy. Yingqian Yin et al. [11] combined black tea image features with hyperspectral features to characterize different grades of black tea. They then created discriminant models using a partial least squares discriminant analysis (PLS-DA), support vector machines (SVMs), and probabilistic neural networks (PNNs), with the SVM model having the highest validation set accuracy (98.33%). Nevertheless, there is a dearth of research currently available that uses fused data to identify soil organic matter. The efficacy of the hyperspectral and machine vision data-fused quantitative soil organic matter prediction model is still being assessed.

The soil of typical Chinese tea plantations is used as the research object in this study, which is based on the aforementioned research state. The soil spectral information and color texture features in the pictures are acquired by using the hyperspectral images that are obtained. To combine the spectral and color texture features of the soil, middle-level data fusion is used. There were three distinct wavelength screening techniques used: VCPA, RF, and VCPA-IRIV. And using the fusion data, the regression models under the SVR and PLSR approaches were created, respectively. From there, a quantitative prediction model of soil organic matter is established to find the best algorithmic strategy, which establishes a theoretical foundation for future advancements in the nondestructive detection technology of soil organic matter.

## 2. Materials and Methods

### 2.1. Collection of Soil Samples and Estimation of the Organic Matter Content

The soil samples utilized in this study were gathered in the Shandong Province locations of Jufeng Town, Rizhao City; Lanshan District, Qingdao City; and Junan County, Linyi City. The six main forms of tea grown in China are cultivated in all three areas: green tea, black tea, white tea, yellow tea, dark tea, and Oolong tea. The sampling areas that were chosen are representative. Three tea gardens provided soil samples. Each tea plantation has five sample locations, designated A, B, C, D, and E, in that order. In all of the municipal tea gardens, 15 separate soil collection locations with a combined sampling area of 25 m^2^ and a depth of 0~20 cm were chosen [12]. Soil samples were then taken ten times from each area for a final total of 150 soil samples.

The gathered samples were taken back to the lab at the Shandong Academy of Agricultural Sciences in Jinan, Shandong Province, China, for the analysis of the organic matter content. To ensure that each soil sample weighed the same weight, contaminants such as stones and animal and plant remains were cleaned before drying. Following that, the criteria for assessing soil organic matter content (NY/T 1121.6-2006) [13] were used to evaluate the organic matter content of all soil samples. Weigh each soil sample that passes through a screen with an aperture of 0.25 mm at 0.05~0.5 g, oxidize the soil’s organic carbon with an excessive potassium dichromate sulfuric acid solution, and then titrate the excess potassium dichromate with a ferrous sulfate standard solution. The quantity of organic carbon was determined by converting the amount of potassium dichromate consumed into organic carbon using the oxidation correction coefficient and then multiplying the result by the constant 1.724 to obtain the amount of organic matter in the soil. The organic matter content of each sampling area is shown in Figure 1. The organic matter content of the soil samples varied due to variations in soil texture.

### 2.2. Hyperspectral Image Acquisition

ISpecHyper-VS1000-Lab hyperspectral (Shenzhen, China) imaging equipment from Lyson Optics was used to gather hyperspectral data. The imaging system consists of an integrated hyperspectral dark box, a hyperspectral camera, a source of light that creates the illusion of sunshine, a diffuse reflectance standard version (3%/50%), and a linear displacement sample stage. The wavelength acquisition range is 300~1000 nm with a spectral resolution of 2.5 nm. Before sampling, the machine was allowed to warm up for 30 min so that the internal equipment performance could settle. After preheating was completed, the camera lens was covered with a lens cap for dark current collection. For consistent tiling, 50 g of each soil sample was weighed and placed in a test dish. The images were placed within the dark box of the hyperspectral image capture system. Before each sample, the system obtains a calibration picture of the reference plate within the dark box. The soil samples were eventually subjected to 150 hyperspectral pictures.

### 2.3. Extraction of Spectral Features and Picture Features

Using the aforementioned collected hyperspectral pictures, the spectral information of the soil samples was retrieved. On the hyperspectral picture of each soil sample, 10 ROI sections were chosen at random. Each ROI is a 1 × 1 cm square area as shown in Figure 2. The spectral data inside the regions are extracted, and the average of the ten sets of data is used as the soil sample’s spectral data. The ENVI 5.3 program was used to carry out this procedure. The collected hyperspectral images were adjusted with the following formula to produce precise reflectance profiles of the samples:(1)$$I_{cor}=\frac{I_{raw}-I_{dark}}{I_{white}-I_{dark}}$$
where *I_cor_* is the hyperspectral image that has been calibrated, *I_raw_* is the picture of the sample that was acquired, *I_dark_* is the image of the dark current that was acquired, and *I_white_* is the image of the diffuse reflective plate.

The image processing program created with the MATLAB GUI module was used to extract the soil’s image characteristics (software copyright number: 2014SR149549). Each hyperspectral image contained 10 randomly chosen zones of interest for the soil, from which the color and texture attributes were retrieved. To produce 9 color features and 5 texture features for each image, the data from the 10 regions of interest of each image were averaged. The red channel mean (R), green channel mean (G), blue channel mean (B), color point mean (H), saturation mean (S), luminance mean (V), supergreen transform (2G-R-B), ratio of the red channel mean to green channel mean (R/G), and color (hab*) are among the color features; the mean gray value (m), standard deviation (δ), consistency (U), entropy (e), and smoothness (r) are among the texture features. The following are the formulae for a few picture features:(2)$$Mean\;gray\;value=\sum_{i}\sum_{j}p(i,j)*i$$
(3)$$Standard\;deviation=\sqrt{\sum_{i}\sum_{j}p(i,j)*{\left(i-Mean\right)}^{2}}$$
(4)$$Consistency=\sqrt{\sum_{i}\sum_{j}p(i,j)*\frac{1}{1+{\left(i-j\right)}^{2}}}$$
(5)$$Entropy=\sum_{i}\sum_{j}p(i,j)*lnp(i,j)$$
where *i* is the row vector in the gray-level matrix, *j* is the column vector in the gray-level matrix, and *p*(*i*, *j*) is the probability of occurrence of gray levels *i* and *j* in the image.

### 2.4. Preprocessing of the Spectral Data and Characteristic Band Screening

The system may be impacted by external conditions during the measurement process. To improve the spectrum features and remove superfluous information such as baseline drift and noise generated during the measurement process, the spectral data are preprocessed. Three preprocessing methods are used, including standard normalized variate (SNV), multisource scattering correction (MSC), and smoothing.

The wavelength range of the hyperspectral imaging equipment employed in this investigation is 300~1000 nm. The wavelength range of the gathered soil spectra is 344~986 nm, with a resolution of less than 2.5 nm. There are 300 wavelength points in all, which is a significant quantity of duplicated data. As a result, distinctive bands were searched for in the gathered spectral data. In this article, three feature band screening techniques, random frog (RF) [14], variable combination population analysis (VCPA) [15], and variable combination population analysis and iterative retained information variable algorithm (VCPA-IRIV) [16], are chosen to filter out irrelevant data and further enhance the model accuracy. Inspired by a frog bouncing on a lotus leaf, the random frog (RF) method is a population-based optimization technique. The procedure of a frog leaping on lotus leaves is simulated by the algorithm to determine the best option. The random frog leaping method is distinguished from other optimization algorithms by its straightforward and understandable iterative updating procedure. Using a continuous contraction of variable space as its foundation, a variable combination population analysis (VCPA) is a hybrid variable selection technique. By repeatedly contracting the variable space, the approach finds the optimal set of variables to increase the model’s predictive capability. The variable combination population analysis and iterative retained information variable algorithm (VCPA-IRIV) incorporates the concept of the IRIV algorithm and is enhanced based on the VCPA algorithm. The mutual information-based feature selection technique known as IRIV is capable of handling the nonlinear interaction between features.

In modeling, data overfitting can occur often, although feature band screening does not always completely prevent it. As a result, further feature extraction from the data is needed using feature extractions. After feature wavelength screening, the principal component analysis (PCA) dimensionality reduction approach was utilized in this work to further reduce the dimensionality of the data. Using mapping, PCA rearranged the data while keeping the primary spectral characteristics and condensing the many feature variables into a handful of key features [17,18].

### 2.5. Data Fusion

To improve the connectivity between several datasets and provide better modeling outcomes, data fusion is used. In this work, the soil’s spectral and color texture data are extracted from the obtained hyperspectral pictures of the soil, and the two separate data groups are combined using a data fusion technique to produce superior prediction outcomes. Low-level fusion, middle-level fusion, and high-level fusion are the three different types of data fusion methodologies. The two data matrices that need to be fused are simply spliced together in low-level fusion. Middle-level fusion is the stitching of the data matrix after performing the characteristic wavelength screening of the spectra. We cannot handle the data with perfect precision while using high-level fusion, and there is a chance that some important information may be lost [19]. As a result, the modeling dataset in this work is processed using the low-level fusion and middle-level fusion approach. The resultant color texture feature matrix is spliced with the spectral data after it has undergone feature wavelength screening and PCA dimensionality reduction, and the newly formed matrix is then normalized to obtain a new data matrix. The newly acquired matrix will be applied to other modeling tasks. The broad idea of data fusion is shown in Figure 3.

### 2.6. Model Building and Evaluation Criteria

The dataset was split into training and validation sets using the Kennard–Stone (K-S) algorithm at a ratio of 3:1 after acquiring the fusion data of soil spectral characteristics and color texture features. Kennard–Stone is an algorithm for sample selection that selects a subset of representative samples from large datasets. The dataset consists of 150 sets of fusion data, including 112 sets of fusion data in the training set and 38 sets in the validation set. A quantitative prediction model for soil organic matter using linear PLSR and nonlinear SVR models was created on the basis of this information. To see how fused data affected the forecast of soil organic matter content, the model of single source data and the model of fused data were compared. Support vector regression (SVR) is a regression approach that utilizes support vector machines as its foundation. It predicts regressions by identifying the best hyperplane inside the feature space. Partial least squares regression (PLSR) is a multivariate regression analysis method, which can be used to build a regression model by downscaling the independent variables and extracting the main components in the presence of multicollinearity between the independent variables.

The correlation coefficient (R_C_), correlation root mean square error (RMSEC), prediction set correlation coefficient (R_P_), prediction root mean square error (RMSEP), and relative percentage of deviation (RPD) were employed in this work as assessment indices for the correction and prediction sets [20]. The more closely the RC and RP approach 1, the more accurate the model is. Similarly, the closer the values of RMESC and RMESP are, the more applicable the model is. If RPD is less than 1.4, the model is subpar and cannot be used for prediction research; if RPD is between 1.4 and 1.8, the model can achieve its prediction objective, but accuracy still needs to be improved; and if RPD is larger than 2, the model has good prediction performance [21].

## 3. Results

### 3.1. Spectral Preprocessing Results

The soil spectral data were collected at 300 wavelength locations between 344 and 986 nm. Both numerically and graphically, the spectral data in the 344–402 nm region exhibit substantial instability. To make the test as accurate as possible, spectral data in this region were removed. Three preprocessing methods, MSC, SNV, and smooth, were used to preprocess the spectral data to compare the modeling effect under different preprocessing methods. The spectrum data were preprocessed using three preprocessing techniques, MSC, SNV, and smooth, to examine the modeling impact under various preprocessing techniques. The spectral profiles following the three preprocesses are shown in Figure 4, together with the soil sample’s original spectral profile.

### 3.2. Characteristic Wavelength Screening Results

The RF, VCPA, and VCPA-IRIV algorithms were used for the screening of the unique wavelengths based on various preprocessing of the spectral data. This enhances the spectral analysis’s accuracy by enabling the selection of representative wavelength characteristics from the raw spectral data. The stochastic selectivity of the variables and the effectiveness of the RF algorithm in selecting spectral feature variables result in a competitive selection mechanism for the variables that in turn ensures their validity [22]. The VCPA technique makes use of the exponentially declining function (EDF), model population analysis (MPA), and binary matrix sampling (BMS), which may effectively account for potential interactions between random variables [23]. A novel approach to variable selection that enables the selective omission of unimportant variables is the VCPA-IRIV method [24]. The number of iterations in the VCPA-IRIV algorithm is set at 50 for the EDF function and 1000 for the BMS function.

The number of wavelengths screened using the three methods after MSC pretreatment was 10, 11, and 43; the number of wavelengths screened using the three methods after SNV pretreatment was 10, 11, and 45; and the number of wavelengths screened using the three methods after smooth pretreatment was 10, 10, and 28. The results of the typical wavelength screening for various preprocessing steps are displayed in Figure 5a–c. It is clear that the predominant concentration of the distinctive wavelengths is in the range of 410 nm, 621~832 nm, and 940 nm. The average spectra of the 15 sampling regions are shown in Figure 5d, and it is clear that there is an obvious absorption peak at 940 nm. This peak is caused by the OH stretching vibration, which creates a slight water vapor absorption band and can be used to characterize the amount of soil organic matter [25]. The primary matching bands of soil organic matter were found to be between 620 and 810 nm in the investigation, and there was only a weak link between soil organic matter and the wavelength at 440 nm [26]. The typical wavelength distribution from this study is consistent.

### 3.3. Predictive Modeling Using Only Spectral Data

Following feature wavelength extraction from the spectral data and PCA downscaling, PLSR and SVR prediction models for soil organic matter were created. Ten sets of main components were utilized as inputs to the models (see fourth column of Table 1), and Table 1 displays the results for each model.

The model effect is enhanced after filtering by characteristic wavelength when compared to the model using raw spectral data. The preprocessing techniques MSC and SNV are generally superior to the smooth procedure, according to a comparison of all models. Additionally, all SVR models outperformed PLSR models, showing how effective nonlinear spectral data modeling was in predicting soil organic matter. The best nonlinear model combination overall had R^2^_P_, RSMEP, and RPD values of 0.973, 0.693, and 6.119, respectively, and was MSC + VCPA-IRIA + SVR. The best linear model combination had R^2^_P_, RSMEP, and RPD values of 0.953, 0.895, and 4.711, respectively, and was SNV + VCPA + PLSR. Figure 6 depicts the connection between these two models’ training and prediction sets. The findings show that it is possible to anticipate the amount of soil organic matter using hyperspectral technology.

### 3.4. Predictive Modeling of Fusion Data

#### 3.4.1. Low-Level Fusion

Low-level model data were first used to create the regression model for soil organic matter. For spectral data, distinctive wavelength extraction is not performed with the low-level fusion data. By splicing the spectral data matrix with the picture feature matrix, the matrix was normalized. The modeling of soil organic matter using SVR and PLSR came next. Table 2 displays the performance metrics for each model of the low-level fusion data.

Low-level fusion data are not well suited for modeling. When compared to models that simply used spectral data, very few models performed better. We then employed a middle-level fusion technique for data fusion to look into the impact of fused data on model performance in more detail.

#### 3.4.2. Middle-Level Fusion

After spectral feature wavelength screening, the spectral matrix and image matrix are stitched together to provide the middle-level fusion data. SVR and PLSR models were created based on the fused data of spectral and image characteristics, respectively, to investigate the predictive modeling impact of the fused data. Table 3 displays the accuracy of each model’s predictions for soil organic matter. A comparison of the data in Table 3 and Table 1 shows that the performance of the models with middle-level fusion data is always better than the performance of the models with single spectral data. With an improvement in R^2^_C_ from 0.901 to 0.980, R^2^_P_ from 0.914 to 0.962, and RPD from 3.456 to 4.563, smooth + VCPA-IRIV + SVR is the set of methods whose performance increase on the model is the largest. In Figure 7b,d, the training-set–prediction-set connections using spectral data and fused data are compared. Figure 7a,c show how spectral data and fused data are optimized for parameters using the smooth + VCPA-IRIV + SVR combination. 

Comparing the modeling performance of the combined data reveals that the best linear model combination is SNV + VCPA + PLSR, with R^2^_C_, R^2^_P_, and RPD values of 0.954, 0.965, and 5.448, respectively, while the best nonlinear model combination is MSC + VCPA-IRIV + SVR, with R^2^_C_, R^2^_P_, and RPD values of 0.995, 0.986, and 8.155, respectively. This is in line with the modeling findings from the spectral data, showing that these two combinations can provide the most accurate predictions of soil organic matter and are thus likely to be used. The model performance results for both combinations are displayed in Figure 8a,b.

## 4. Discussion

### 4.1. Analysis of Data Fusion Model Results

In this paper, low-level fusion data and middle-level fusion data were used to construct a prediction model for soil organic matter content. The findings indicate that low-level fused data do not exhibit strong model performance. This shows that the two data matrices are not combined to their full potential by just splicing them together. The spectral data in the middle-level fusion were screened for characteristic wavelengths. By eliminating unnecessary wavelengths, this process enhances the performance of the model. Thus, it is advantageous to combine data from several sources utilizing an intermediate fusion measurement technique in order to produce a superior regression model.

### 4.2. Analysis of the Impact of Each Algorithm on the Results

In this study, the SVR model performs better than the PLSR model for every model. This implies that when it comes to predictive modeling of soil organic matter, nonlinear regression approaches are more advantageous. Compared to the other three preprocessing techniques, the smooth algorithm performs somewhat worse. This is so because the smooth method does not actually remove the effects of scattering in the spectrum; it only smooths the spectrum. However, the scattering effect in the spectrum may be removed by both MSC and SNV.

In the final intermediate fusion algorithm, the VCPA series performs better. It shows that the VCPA series is more applicable in this study. Among VCPA series of algorithms, the VCPA-IRIV algorithm performs better than the VCPA algorithm. The IRIV method incorporated in the VCPA-IRIV algorithm selects wavelengths with stronger correlation.

## 5. Conclusions

In this work, the SVR and PLSR models were built using the low-level fusion and middle-level fusion approach to combine the soil hyperspectral and picture texture characteristic data. The results of the study show that low-level fusion does not produce good results. The middle-level fusion strategy is a better choice. All models with middle-level fusion data are superior to models using only spectral data. Out of the three, the nonlinear model performs better at estimating the amount of soil organic matter. The most accurate SVR model is the one built using the MSC and VCPA-IRIV algorithms. It had an R^2^_C_ of 0.995, an R^2^_P_ of 0.986, and an RPD of 8.155. 

The results of this study demonstrate the feasibility of soil organic matter content prediction with fusion data, which improves the accuracy of the prediction model. It is beneficial to promote accurate control of soil organic matter content in tea plantations. The use of organic matter can increase soil fertility and encourage the growth of tea trees because it can decompose into nutrients that plants require. Precision irrigation can be achieved based on the need of tea trees for organic matter content in the soil. This promotes the growth and development of the tea trees.

## Figures and Tables

**Figure 1 sensors-23-09684-f001:**
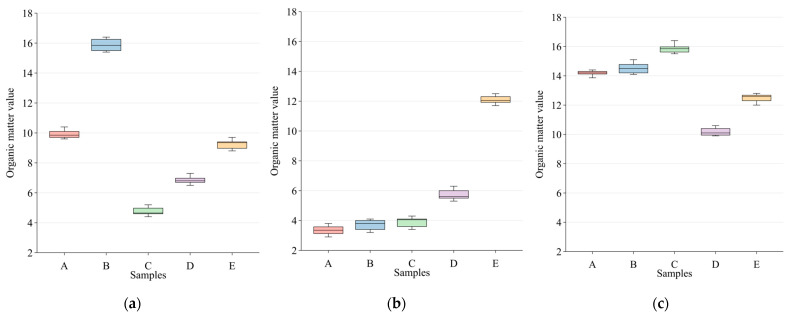
Organic matter content of soil samples from three municipalities(five sampling sites were delineated as A, B, C, D, and E in the tea gardens of each city). (**a**) Soil samples of tea plantations in Rizhao; (**b**) Soil samples of tea plantations in Qingdao; (**c**) Soil samples of tea plantations in Linyi.

**Figure 2 sensors-23-09684-f002:**
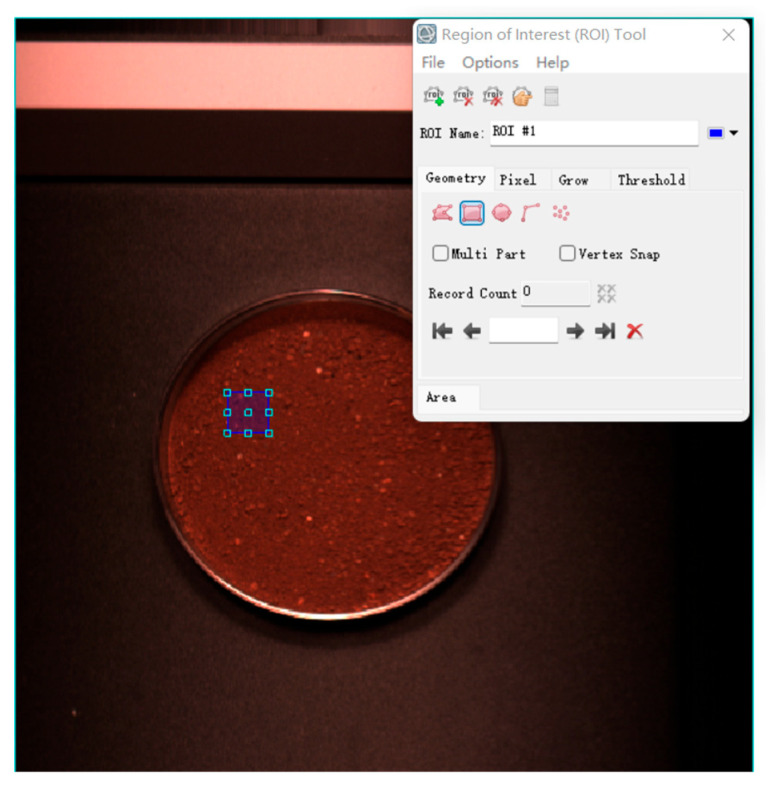
Schematic diagram of ROI region selection.

**Figure 3 sensors-23-09684-f003:**
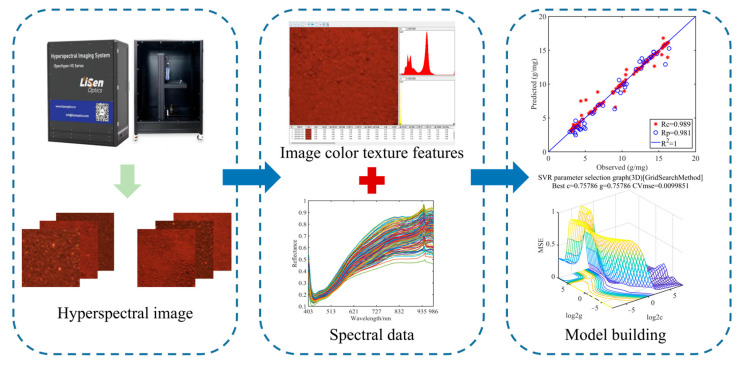
Flowchart of data fusion concepts.

**Figure 4 sensors-23-09684-f004:**
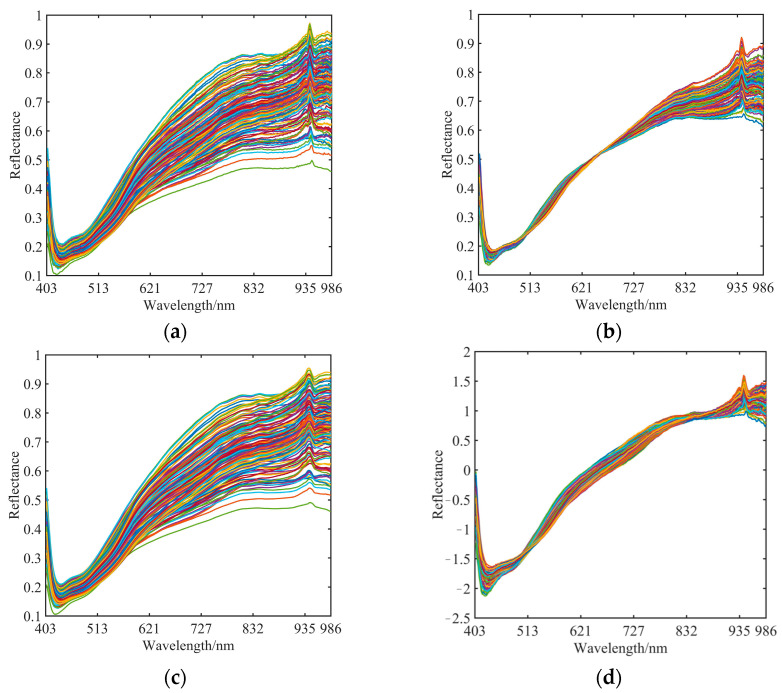
Raw spectra and various preprocessed spectra. (**a**) Raw spectra; (**b**) Spectra after MSC processing; (**c**) Spectra after smooth processing; (**d**) Spectra after SNV processing.

**Figure 5 sensors-23-09684-f005:**
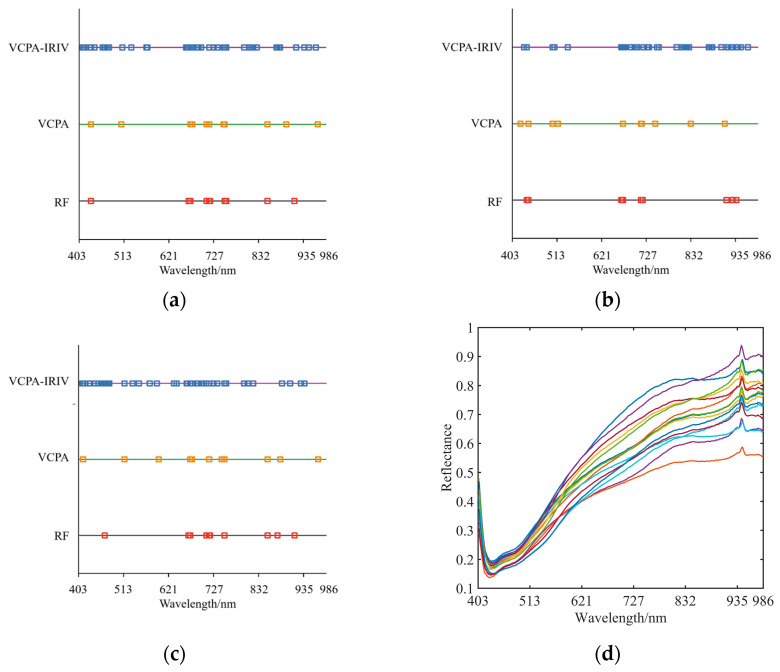
Characteristic wavelength screening results and average spectra of 15 sample points. (**a**) Distribution of characteristic wavelength screening after MSC preprocessing; (**b**) Distribution of characteristic wavelength screening after smooth preprocessing; (**c**) Distribution of characteristic wavelength screening after SNV preprocessing; (**d**) Average spectra of 15 sample points.

**Figure 6 sensors-23-09684-f006:**
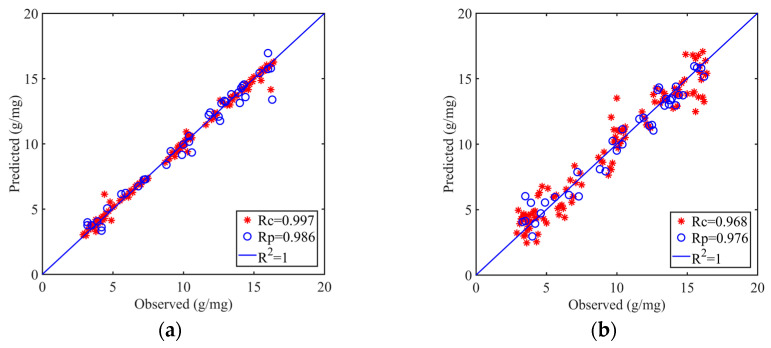
Comparison of model training set and prediction set under single spectral data. (**a**) MSC + VCPA-IRIA + SVR; (**b**) SNV + VCPA + PLSR.

**Figure 7 sensors-23-09684-f007:**
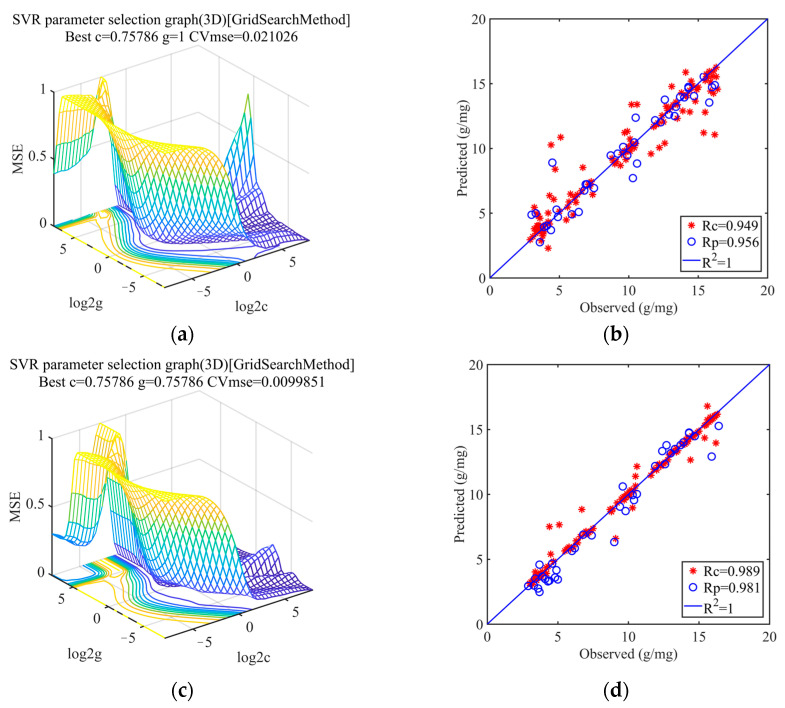
Comparison of non-fusion and fusion data models under smooth + VCPA-IRIV approach. (**a**) Parameter optimization process for a single spectral data model; (**b**) Comparison of training and prediction sets of single spectral data models; (**c**) Parameter optimization process for fusion data models; (**d**) Comparison of training and prediction sets of fusion data models.

**Figure 8 sensors-23-09684-f008:**
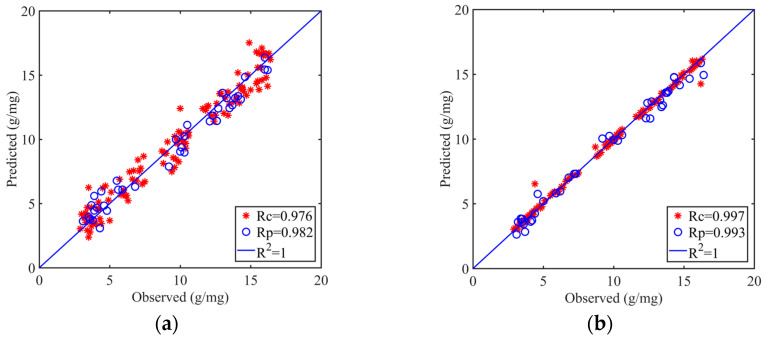
Effectiveness of linear and nonlinear optimal models under fusion data. (**a**) SNV + VCPA + PLSR; (**b**) MSC + VCPA-IRIV + SVR.

**Table 1 sensors-23-09684-t001:** Predictive modeling results for spectral data.

Preprocessing Methods	Data Dimensionality Reduction	Models	PCs	Calibration Set		Prediction Set		RPD
				R^2^_C_	RMSEC	R^2^_P_	RMSEP	
MSC	None	SVR	5	0.922	1.268	0.928	1.097	3.696
		PLSR	8	0.877	1.585	0.895	1.265	3.134
	RF	SVR	5	0.978	0.677	0.957	0.822	4.847
		PLSR	9	0.868	1.658	0.799	1.766	2.261
	VCPA	SVR	5	0.972	0.756	0.943	0.672	5.981
		PLSR	7	0.925	1.233	0.950	0.887	4.533
	VCPA-IRIV	SVR	7	0.994	0.350	0.973	0.693	6.119
		PLSR	10	0.926	1.234	0.892	1.205	3.083
SNV	None	SVR	6	0.946	1.061	0.933	0.950	3.667
		PLSR	10	0.848	1.785	0.822	1.644	2.406
	RF	SVR	4	0.901	1.419	0.914	1.219	3.456
		PLSR	7	0.859	1.663	0.885	1.443	2.999
	VCPA	SVR	7	0.965	0.846	0.964	0.807	5.273
		PLSR	9	0.937	1.118	0.953	0.895	4.711
	VCPA-IRIV	SVR	6	0.984	0.565	0.960	0.768	5.071
		PLSR	9	0.901	1.401	0.909	1.200	3.376
Smooth	None	SVR	5	0.904	1.412	0.901	1.292	3.188
		PLSR	10	0.896	1.475	0.803	1.730	2.287
	RF	SVR	4	0.960	0.895	0.935	1.079	3.940
		PLSR	8	0.827	1.854	0.761	2.100	2.076
	VCPA	SVR	5	0.926	1.213	0.909	1.337	3.345
		PLSR	9	0.946	1.017	0.940	1.082	4.168
	VCPA-IRIV	SVR	4	0.901	1.419	0.914	1.219	3.456
		PLSR	10	0.921	1.267	0.907	1.258	3.335

**Table 2 sensors-23-09684-t002:** Performance metrics for each model with low-level fusion data.

Preprocessing Methods	Models	Calibration Set		Prediction Set		RPD
		R^2^_C_	RMSEC	R^2^_P_	RMSEP	
MSC	SVR	0.986	0.528	0.941	0.970	3.960
	PLSR	0.882	1.565	0.831	1.557	2.468
SNV	SVR	0.986	0.272	0.950	0.825	4.706
	PLSR	0.932	1.178	0.895	1.263	3.141
Smooth	SVR	0.923	1.290	0.923	1.237	3.382
	PLSR	0.874	1.523	0.882	1.460	2.915

**Table 3 sensors-23-09684-t003:** Performance results of each model for fusion data.

Preprocessing Methods	Data Dimensionality Reduction	Models	PCs	Calibration Set		Prediction Set		RPD
				R^2^_C_	RMSEC	R^2^_P_	RMSEP	
MSC	RF	SVR	9	0.990	0.622	0.959	0.948	4.938
		PLSR	9	0.888	1.442	0.889	1.547	3.051
	VCPA	SVR	9	0.983	0.579	0.976	0.660	6.459
		PLSR	10	0.951	0.987	0.950	0.941	4.534
	VCPA-IRIV	SVR	10	0.995	0.312	0.986	0.558	8.155
		PLSR	10	0.947	1.020	0.921	1.252	3.600
SNV	RF	SVR	9	0.989	0.480	0.962	0.851	5.008
		PLSR	10	0.912	1.327	0.923	1.191	3.650
	VCPA	SVR	9	0.995	0.323	0.970	0.761	5.854
		PLSR	10	0.954	0.956	0.965	0.818	5.448
	VCPA-IRIV	SVR	9	0.992	0.406	0.982	0.639	6.957
		PLSR	10	0.903	1.373	0.925	1.233	3.704
Smooth	RF	SVR	8	0.981	0.623	0.950	0.942	4.530
		PLSR	10	0.894	1.442	0.904	1.306	3.267
	VCPA	SVR	8	0.976	0.710	0.942	0.972	4.051
		PLSR	10	0.965	0.831	0.941	0.988	4.156
	VCPA-IRIV	SVR	8	0.980	0.639	0.962	0.951	4.563
		PLSR	10	0.940	1.095	0.925	1.132	3.693

## Data Availability

The data presented in this study are available on request from the corresponding author.

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
