# Peer review of "Research on the Detection Method of Organic Matter in Tea Garden Soil Based on Image Information and Hyperspectral Data Fusion"

_sensors, 2023, doi:10.3390/s23249684_

Round 1

Reviewer 1 Report

Comments and Suggestions for Authors

s. attached PDF file

Comments on the Quality of English Language

Reviewer 2 Report

Comments and Suggestions for Authors

In this manuscript, the authors described “Research on the Detection Method of Organic Matter in Tea Garden Soil based on Image Information and Hyperspectral Data Fusion”. However, there are some problems in the study, here are some points of improvement:

Introduction:

1. The detailed background on soil organic matter is commendable, but after establishing the importance of soil organic matter, a concise review of earlier methodologies for this purpose would be enriching. In particular, in lines 33-34, the author should describe in detail the traditional techniques for determining soil organic matter, and indicate the shortcomings of these techniques to better reflect the advantages of hyperspectral imaging techniques for soil detection.

2. The author should briefly introduce hyperspectral imaging technology in the introduction so that readers can better understand its application in different industries.

3. A brief overview of the algorithms and methods involved in the methodology of this study is lacking.

Materials and Methods:

1. Whether the method of collecting soil samples is scientific, it is suggested to cite relevant literature to support it. What makes the chosen regions unique?

2. What do the different colors in Figure 1 represent.

3. The author should list the formula for obtaining features, for example, the formula for calculating entropy in texture features.

4. The author should briefly introduce the band screening and modelling methods used in the paper.

Results and discussion:

1. “Among these, low-level fusion merely splices the two data matrices and does not yield superior outcomes.” The author should list the low-level fusion results in the text to make them more persuasive.

2. It's crucial to discuss not just the variation in different models but the reasons behind them.

3. You did not develop models based on raw spectral data. Why?

4. Lines 247-248 show SMOOTH+VCPA-IRIV+SVR is the combination of modeling approaches that has the greatest change in impact. However, the results obtained by this method are not the best, which is inferior to the results of SNV+VCPA+SVR. Why choose SMOOTH+VCPA-IRIV+SVR?

5. Discussion should be expanded comparing your results with the existed literature.

Conclusions

1.Your conclusion succinctly recaps the research findings. However, a more encompassing view of the results and their broader implications would elevate it. Discuss the larger benefits of this technology for various stakeholders. Address any encountered challenges and future strategies to overcome them. A powerful closing remark, emphasizing the research's overarching significance, can leave a lasting impact.

Comments on the Quality of English Language

Moderate editing of English language required

Round 2

Reviewer 2 Report

Comments and Suggestions for Authors

Accept in present form.